# Insights into the State of the Art of Urogenital Schistosomiasis with a Focus on Infertility

**DOI:** 10.3390/tropicalmed9080177

**Published:** 2024-08-10

**Authors:** Rafaella P. Marques, Waqas Ahmad, Raquel Soares, Katia C. Oliveira, Monica C. Botelho

**Affiliations:** 1Disciplina de Parasitologia, Departamento de Microbiologia, Imunologia e Parasitologia, Escola Paulista de Medicina, Universidade Federal de São Paulo, São Paulo 04023-062, Brazil; rafaella.marques@unifesp.br (R.P.M.); katia.oliveira@unifesp.br (K.C.O.); 2Department of Clinical Sciences, University of Veterinary and Animal Sciences (UVAS), KBCMA Campus, Narowal 51800, Pakistan; waqas.hussain@uvas.edu.pk; 3Department of Biomedicine, Faculty of Medicine, University of Porto, 4200-450 Porto, Portugal; raqsoa@med.up.pt; 4i3S—Instituto de Investigação e Inovação em Saúde, Universidade do Porto, 4200-135 Porto, Portugal; 5Department of Health Promotion and Chronic Diseases, INSA—National Institute of Health Dr. Ricardo Jorge, 4000-055 Porto, Portugal

**Keywords:** schistosomiasis, urogenital schistosomiasis, infertility, catechol estrogen imbalance

## Abstract

Schistosomiasis is a neglected tropical disease that affects developing countries worldwide and is caused by several species of parasites from the *Schistosoma* genus. Chronic infection is characterized by the formation of granulomas around the parasite eggs, the leading cause of pathology. The hepatosplenic clinical form is one of the most common, but urogenital schistosomiasis is another relevant clinical presentation responsible for infertility in men and women. Inflammatory response, anatomical deformations, and endocrine/biochemical changes are involved in the development of infertility. Schistosome parasites can synthesize catechol estrogen-like molecules and affect the sexual hormone balance in their host. Here, we review many aspects of the pathology of urogenital schistosomiasis, specifically infertility, and point to the biochemical and endocrinal elements that must be investigated in the future.

## 1. Introduction

As described by the German surgeon Theodor Bilharz in 1851, schistosomiasis (also known as bilharziasis) is an important parasitic disease worldwide. This helminthiasis is endemic in areas of developing countries. The highest risk of contamination resides in populations that live in areas with the absence of basic sanitation and depend on agriculture and fishing for their subsistence [1,2,3] The disability-adjusted life year of schistosomiasis in 2019 (1.6 million) shows a 24% reduction in comparison to 2.2 million in the year 2000, concomitant with the increase in worldwide coverage of the control program [4]. Human schistosomiasis is caused by trematode flukes belonging to the genus *Schistosoma*: *S. mansoni*, *S. haematobium*, *S. japonicum*, *S. malayensis*, *S. intercalatum*, and *S. mekongi*, and infects people in the endemic areas of Central and South America, the Caribbean, Middle and East Asia, and Africa [5,6,7].

The life cycle of schistosomes is established with the blood flukes (schistosomes) laying eggs in the veins of the mesenteric, bladder or reproductive organ plexuses. Their eggs are excreted from the human body through urine or feces into fresh water, and the larvae (miracidia) are liberated. The miracidia swim to their intermediate host, a snail (there is a specific species for each schistosome species), and penetrate them. After several weeks of asexual reproduction and growth, the infected snail releases (under environmental stimuli) another larval form known as cercariae. These cercariae swim until they locate a human host, at which point they penetrate the intact skin. Once inside the human host, these cercariae shed their tails and transform into schistosomula. These schistosomula then develop a resilient tegument that shields them from immune responses. They migrate into the circulation and, around 30 days later, they mature into adult worms within the portal vein. These adult worms proceed to migrate within the venous system (around 45 days), taking residence in the small vessels of the intestines (*S. mansoni* and *S. japonicum*), bladder or reproductive organs (*S. haematobium*), according to the species. Adult worms live in the blood vessels for years, excreting hundreds of eggs daily and successfully surviving in the immune system with sophisticated evasion mechanisms. The eggs must either exit the body through excretion (feces or urine) or become ensnared within adjacent tissues. When eggs become entrapped, they stimulate the development of periovular granulomas, the key event of the physiopathology of schistosomiasis, and lead to a range of consequences depending on parasite load, age, and immune/physiological status of the host. These effects encompass physical and intellectual development limitations and influence the endocrine and immunological systems (by producing hormone-like molecules) and their signaling molecules (such as receptors, kinases and transcription factors). Additionally, organ-specific outcomes emerge, including pronounced hepatosplenomegaly, portal fibrosis, and, as a result, portal hypertension [2,7,8].

To comprehensively understand the pathology of urogenital schistosomiasis and its impact on fertility, our group has been investigating the biochemical and endocrine elements of this disease. Our previous research in this area explained some of the mechanisms by which schistosome parasites influence hormonal balance and reproductive function in the host. Additionally, the elucidation of these mechanisms may pave the way for the development of targeted interventions to prevent or treat infertility associated with urogenital schistosomiasis. Urogenital schistosomiasis represents a significant public health concern, particularly due to its association with infertility. Continued research into the biochemical and endocrine aspects of the disease is crucial for improving our understanding of its pathogenesis, especially infertility, and developing effective strategies for prevention and treatment.

Here, we review these aspects based on the search of the literature deposited in the PubMed database. We included all publications that reported schistosomiasis associated with infertility. The criteria and search strategies used for the inclusion of the papers are elucidated in Appendix A).

## 2. Urogenital Schistosomiasis

Eggs cause inflammation in the urinary and genital tracts, intestinal walls, and liver. The pathological effects in the genital tract lead to obstruction, fibrosis, the destruction of anatomical structures, and the production of antibodies against spermatozoa [9]. Although the development of genital tract pathologies can occur in both genders, women are more affected. *S. haematobium* eggs can be deposited in the urinary bladder, lower ureters, cervix, and vagina, where the main pathology occurs. The close location of the genital venous plexus favors the parasitic migration, leading to the development of female genital schistosomiasis (FGS). FGS outcomes derived from the deposition of schistosome eggs include the formation of granulomas, fibrosis, and angiogenesis as part of the inflammatory response. Consequently, women afflicted by this condition might undergo symptoms such as hematuria, dysuria, and an elevated susceptibility to the development of bladder cancer [8,10,11,12,13,14].

The classical lesions of female genital schistosomiasis are described as “sandy patches” because they comprise eggs and granulomas associated with increased inflammation and vascularization. Typically, areas of sandy patches tend to bleed upon contact and can additionally induce genital itching and discomfort. This condition can also cause stress incontinence, dyspareunia, and potentially even infertility [15]. Endometriosis can be linked to FGS, often contributing to female infertility and an elevated risk of the horizontal transmission of the human immunodeficiency virus (HIV) [2,13,16].

Current guidelines recommend preventive chemotherapy, using praziquantel to avoid the impact of schistosomiasis on life quality. A single dose of praziquantel in participants in a study in Nigeria significantly reduced eggs in their urine [17]. Urogenital schistosomiasis maintains a significant prevalence, mainly affecting female school aged children (SAC) and often goes undetected among individuals [18,19]. The infection with *S. haematobium* can be detected by testing aspects like proteinuria and hematuria, but the progression of the disease and complications need more sophisticated approaches (such as ultrasound), which impacts the capability to detect the disease in remote areas [20,21]. Prevention is the most effective way to decrease the disease’s occurrence [2,13,16].

## 3. Mechanisms Underlying Schistosomiasis-Associated Infertility

Infertility is characterized as the inability to achieve pregnancy following 12 months of frequent and unprotected sexual intercourse. This health condition impacts one in six couples (15–20%) globally and 12–30% of couples in the sub-Saharan Africa region [22,23,24]. Pelvic infections (such as *Neisseria gonorrhea* and *Chlamydia trachomatis*) are frequently associated as a potential cause of infertility. Schistosomiasis persists as a significant risk factor, since many cases of infertility in schistosomiasis-endemic areas have been connected to the clinical presentation on female genitalia (41%) and are involved in many cases of ectopic pregnancies (3.6%) [9,22,25].

### 3.1. Periovular Granuloma Is the Critical Event of Urogenital Pathogenesis

Infertility is thought to be associated with schistosome egg deposition, which triggers inflammation and granulomatous reactions through the release of proteolytic enzymes by the eggs. This process can lead to mechanical blockages, the formation of scar tissue, and the destruction of anatomical structures [9,26]. The granuloma formed around the eggs accounts for this, comprising eosinophils, mononuclear phagocytes, fibroblasts, lymphocytes, neutrophils, plasma cells, and mast cells. The presence of parasites in organisms triggers the systemic immune response. Antigens originating from cercariae, adult worms, and eggs are transported to lymphoid organs, which are taken up, processed, and presented to antigen-specific T cells. This sequence of events triggers T-cell activation and the secretion of lymphokines. Consequently, systemic immune responses are activated, which can lead to chronic fibrosis and tubal adhesions that significantly contribute to infertility [27]. The lesions caused by the presence of eggs in the lower genital tract, especially in the vagina or vulva, can become hypertrophic, growing for months or even years. Consequently, the weakening of epithelial barriers due to vaginal and cervical lesions can facilitate HIV transmission. Easy bleeding in these areas allows HIV in semen direct access to the bloodstream and, through ulcerative lesions, to the regional lymph nodes [27,28]. The diagnosis of lesions in the upper genital tract, which includes the uterus, fallopian tubes, and ovaries, is more difficult in routine clinical practice [27]. It is worth highlighting that infertility and reduced fertility can also be attributed to hormonal imbalances [27,28].

Male infertility can be caused by various conditions, like sexually transmitted infections, febrile diseases, and parasitic infections, such as filariasis and schistosomiasis. The association between male infertility and schistosomiasis is little reported, and this pathology can occur by a range of mechanisms: (i) hypothalamic–pituitary–gonadal axis hormonal disruption; (ii) testicular tissue damaged by inflammation and granuloma formation and/or (iii) the obstruction of or an accessory sex organ affected by inflammation and granuloma formation leading to severe oligozoospermia, azoospermia, or subfertile semen [26]. Trapped Schistosoma eggs in men can block the spermatic venous plexus, leading to granuloma formation, testicular infarction, intense epididymitis, and the inhibition of spermatogenesis [3,29]. Seminal vesiculitis can result in hematospermia, painful ejaculation, burning during urination, and lower back pain; the granuloma stage causes irreversible damage [3,29].

Schistosomiasis primarily involves thromboembolic lesions leading to terminal vein occlusion and blood supply impairment [30]. The access routes for the worms in the venous systems of the spermatic cord, epididymis, and testicle include the following: the junctions of the superior mesenteric with the right spermatic veins and the inferior mesenteric with the left spermatic veins; the convergence of the deferential vein with pelvic veins; the linkage between the pampiniform plexus in the scrotum and pelvic veins intersecting near the external inguinal ring; and the possibility of direct spread from the epididymis to the testis (though rare) and the potential for systemic arterial spread, as observed in cases of ectopic bilharzial lesions (although also rare) [31]. Figure 1 illustrates the principal events associated with female and male infertility in urogenital schistosomiasis.

### 3.2. Estrogen Pathways and Catechol Estrogens: New Metabolites Involved in Urogenital Schistosomiasis-Induced Bladder Cancer

Estrogen signaling is mediated through estrogen receptors (ERs), including the classical receptors ERα and ERβ, as well as membrane receptors such as GPR30 [32,33]. Estrogen receptor α is predominantly expressed in gonadal organs, the liver, and adipose tissue, while ERβ is primarily found in the prostate, bladder, ovary, colon, adipose tissue, and immune system [34]. These classic receptors play essential physiological roles in ovarian development and function and mediate estrogen’s cardioprotective effects [32,35]. Previous studies have linked estrogen-like metabolites during female genital schistosomiasis (FGS) with self-reported infertility. Santos et al. reported that these metabolites down-regulate ERα and ERβ in estrogen-responsive cells in vitro [36]. Different estrogen signaling pathways are related to the regulation of biological processes. In the classical pathway, activating the ligand and direct binding to Estrogen Responsive Elements (EREs) in the promoter regions of target genes results in transcriptional alterations in estrogen-responsive genes. As for the non-genomic pathway, ligand-activated ERs do not directly bind to DNA. Instead, they are connected through the activation of a kinase cascade to the activation of a transcription factor complex, which subsequently interacts with the target gene promoter. Then, gene regulation is affected by indirect DNA binding [32,37,38,39].

Catechol estrogens are the result of the metabolism of estrogens E1 and E2. In this process, the formation of pregnenolone occurs after the catalysis of cholesterol by the CYP11A enzyme of the cytochrome P450 family. Subsequently, another P450 enzyme, CYP17A, leads to the formation of androstenedione, which can be catalyzed by HSD17B3 for the formation of testosterone or metabolized by aromatase (CYP19A), resulting in estrone (E1) production. Estradiol (E2) formation occurs through the aromatization of testosterone or the metabolization of estrone by HSD17B1. The hydroxylation of the steroid aromatic ring on C2, C4, and C16 positions by cytochrome P450 enzymes, such as CYP1A1, CYP1A2, CYP3A4, and CYP1B1, culminates in the production of catechol estrogens 4-hydroxyestrone and 4-hydroxyestradiol (or 2-hydroxyestrone and 2-hydroxyestradiol), which CYP450 oxidates in the semiquinones E1-3,4-SQ and E2-3,4-SQ (or E1-2,3-SQ and E2-2,3-SQ) and the posterior to quinones E1-3,4-Q and E2-3,4-Q (or E1-2,3-Q and E2-2,3-Q). Evidence has demonstrated that *S. haematobium* eggs secrete these novel catechol estrogens which are then metabolized into active quinones capable of modifying DNA, either by forming adducts or by producing reactive oxygen species, in addition to inducing mutations in the p53 gene, resulting in damage and carcinogenesis [36]. Consequently, it is assumed that parasite-derived catechol estrogens may impact infertility during *Schistosoma* infection by interfering with estrogen receptors [36,40,41,42].

Figure 2 postulates the catechol estrogens biosynthesis pathway in *S. haematobium*, *S. mansoni*, and *S. japonicum* based on a brief in silico search for homologs in the genome of these parasites. The result of this search is documented in Table 1. There, we point to the gene predictions that may be responsible for the catechol estrogen synthesis. The hypothesis is that the synthesis of these hormone-like molecules by parasites interferes with the fertility and/or pathological process of cancer development, for example.

The crosstalk between hosts and parasites, particularly in the context of hormonal changes induced by parasites, presents a fascinating area for further research. This may be the effect of the parasite’s metabolic interference in the substrates of the host, interfering in the endocrinal regulation of sexual hormones or producing hormone-like molecules that may act as endocrinal disruptors. Investigating the impact of parasites on host hormonal regulation, especially regarding fertility-related hormones, could provide valuable insights into the mechanisms of parasitic infections and their effects on reproductive health.

Examining the genital organs of infected animals could shed light on potential alterations induced by the parasite’s production of catechol estrogens in the reproductive system. Similarly, exploring the synthesis of estrogen-related metabolites by parasites, such as eggs, and their interaction with estrogen receptors (ER) in host cells, would be crucial for revealing hormonal interactions and/or hormonal alterations in infected hosts. Understanding whether the levels of sexual hormones in infected female hosts influence the expression of ER in their tissues is another important aspect to explore. This could elucidate how parasitic infections impact the hormonal balance and reproductive physiology of the host. Additionally, investigating whether parasites interfere with the physiological processes of fertility could provide insights into potential mechanisms underlying infertility associated with schistosomiasis.

## 4. Conclusions and Future Perspectives

The established molecular crosstalk between the host and parasite is complex and exciting. Further research is required to examine the impact of hormonal changes induced by the parasite in the host. It would be interesting, for example, if the genital organs of infected animals were inspected. This research would also include an investigation of the synthesis of estrogen-related metabolites by parasites (eggs) that can down- or up-regulate the expression of estrogen receptor (ER) alpha and beta in estrogen-responsive cells. Could the level of sexual hormones in infected host females induce changes in the ER expression in their tissues? Could the parasite interfere with the physiological process of fertility? All these points are relevant to understanding the impact of schistosomiasis on the fertility status of couples and the molecular mechanisms beyond it. Overall, unraveling the molecular mechanisms underlying the interaction between schistosomiasis and fertility could have significant implications for understanding the reproductive health implications of parasitic infections. This research could contribute to the development of targeted interventions to mitigate the impact of schistosomiasis on fertility status and improve reproductive health outcomes for affected individuals and couples.

## Figures and Tables

**Figure 1 tropicalmed-09-00177-f001:**
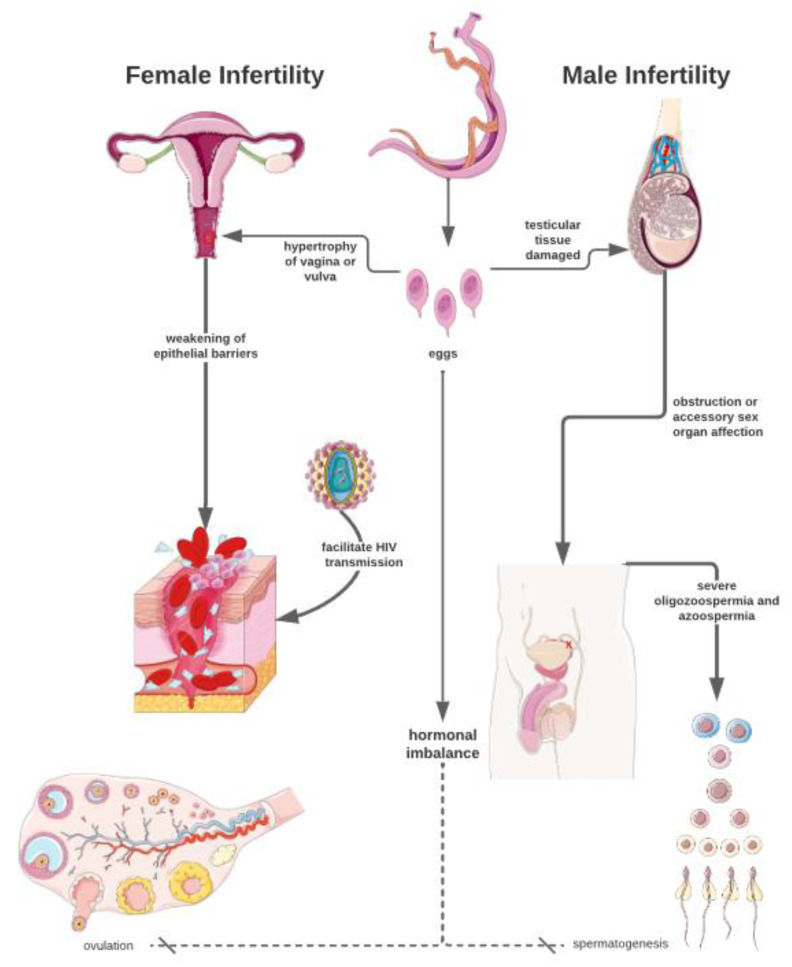
Events associated with the female and male infertility in urogenital schistosomiasis.

**Figure 2 tropicalmed-09-00177-f002:**
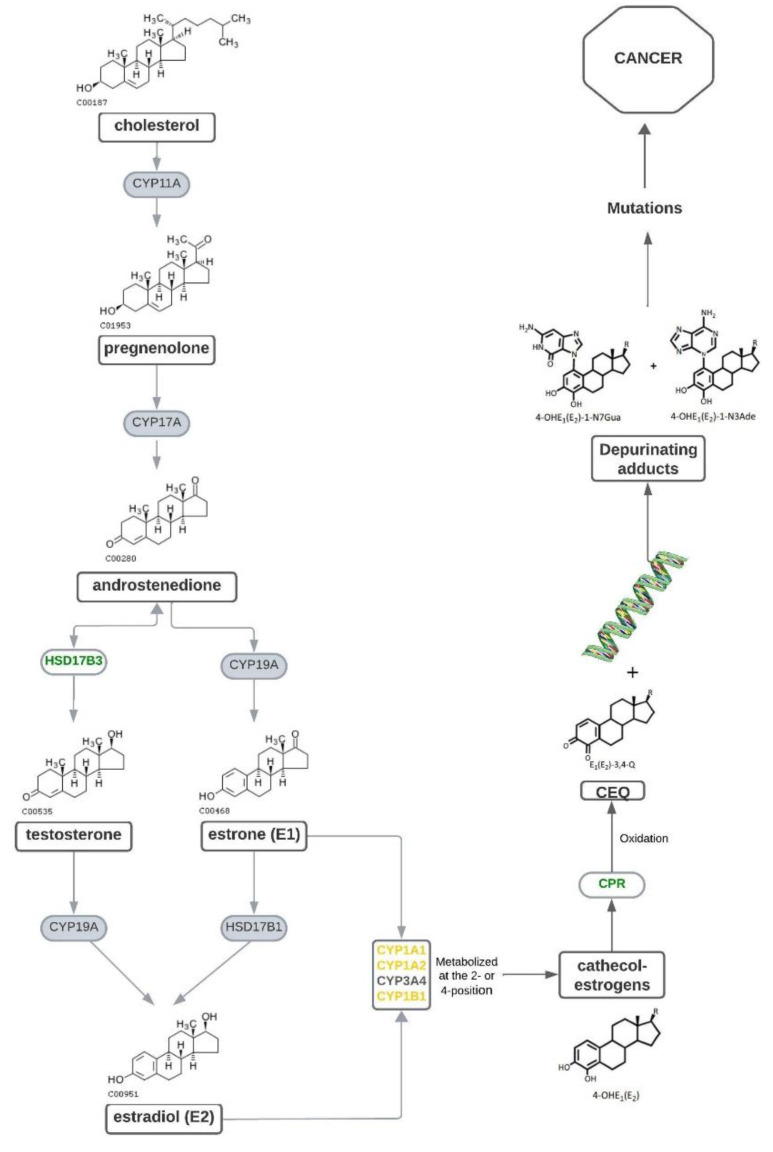
Catechol estrogens synthesis pathway in humans and the possible homologs identified in schistosome species. In green are highlighted the possible homologs and in yellow are highlighted the homologs with an e-value close to the cut-off (10^−5^).

**Table 1 tropicalmed-09-00177-t001:** *S. haematobium*, *S. mansoni*, and *S. japonicum* homologs of postulates catechol estrogens biosynthesis pathway.

*S. japonicum*			*S. mansoni*			*S. haematobium*			Human Ref. Seq. ID.	Name	Symbol	Enzyme ID
% Identity	E-Value	Gene ID	% Identity	E-Value	Gene ID	% Identity	E-Value	Gene ID				
–	–	–	–	–	–	–	–	–	ALQ33469.1	cholesterol monooxygenase	CYP11A	1.14.15.6
–	–	–	–	–	–	–	–	–	ABY87534.1	steroid 17alpha-monooxygenase	CYP17A	1.14.14.19
53	1.1 × 10^22^	Sjp_0058300	52.5	7.40 × 10^−21^	Smp_009430.1	53.8	2.20 × 10^−23^	SHAE2_97190.1	EAW92645.1	testosterone 17beta-dehydrogenase	HSD17B3	1.1.1.64
–	–	–	–	–	–	–	–	–	NP_000404.2	hydroxysteroid 17-beta dehydrogenase 1	HSD17B1	1.1.1.62
–	–	–	–	–	–	–	–	–	NP_059488.2	cytochrome P450 family 3 subfamily A member 4	CYP3A4	1.14.13.32
37.9	5.00 × 10^−8^	EWB00_005315.2	30.2	1.30 × 10^−6^	Smp_156400.1	30.2	3.50 × 10^−7^	MS3_00006533.2_mrna	NP_001306146.1	cytochrome P450 family 1 subfamily A member 1	CYP1A1	1.14.14.1
34.5	1.00 × 10^−6^	EWB00_005315.2	27	2.50 × 10^−5^	Smp_156400.1	27	5.00 × 10^−6^	SHAE1_52460.1	NP_000752.2	cytochrome P450 family 1 subfamily A2	CYP1A2	1.14.14.1
36.4	5.00 × 10^−5^	EWB00_005315.3	35.7	3.60 × 10^−5^	Smp_156400.1	33.9	9.60 × 10^−5^	SHAE2_54450.1	NP_000095.2	cytochrome P450 family 1 subfamily B1	CYP1B1	1.14.14.1
43.9	1.20 × 10^−64^	EWB00_007185	42.6	1.60 × 10^−64^	Smp_030760.1	43.4	4.20 × 10^−64^	SHAE1_77650.8	NP_000095.2	NADPH--cytochrome P450 reductase	CPR	1.6.2.4

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
