# Peer review of "Insights into the State of the Art of Urogenital Schistosomiasis with a Focus on Infertility"

_tropicalmed, 2024, doi:10.3390/tropicalmed9080177_

Round 1
Reviewer 1 Report
Comments and Suggestions for Authors
Dear authors,
Thank you for this notable research work. Below are the comments and suggestions on the research paper entitled " Insights into the state of the art of urogenital schistosomiasis".
The manuscript is well written and has presented significant points on the pathology of schistosomiasis. However, the following comments must be addressed to further improve the research paper:.
It is suggested to improve the figures on the research paper by placing a photo with good resolution and visible text.
It is also suggested to add more related literature to the research paper on studies associating infertility with schistosomiasis.
The scientific name S. haematobium must be italicized in lines 196 and 203.
These are some of the suggestions to substantiate the relevance of this research paper. Kindly highlight in red the revisions made to the revised manuscript for confirmation upon submission of the revised paper.
Thank you for your continued research work.
Author Response
Please find our reply in the attached file.

Reviewer 2 Report
Comments and Suggestions for Authors
This is a very nice topic and have an enormous impact in the understanding of schisto-induced uro-genital pathologies. However, to make the story following aspects must be addressed-
1. In introduction pls highlight more about the burden of the disease, including mention DALY
2. To make the story complete, pls provide the diagnosis, treatment and present on-going control programs
3. Please mention about the selection of article: criteria for exclusion and inclusion of literatures along with a flow diagram
4. references are too old; there are a lot of recent publication, some are as follows-
Percheron L, Leblanc C, Ulinski T, Fila M, Malvy D, Bacchetta J, Guigonis V, Debuisson C, Launay E, Martinez E, Morand A, Decramer S, Schanstra JP, Berry A. Pediatric urogenital schistosomiasis diagnosed in France. Pediatr Nephrol. 2024 Jun;39(6):1893-1900. doi: 10.1007/s00467-023-06260-x. Epub 2024 Jan 12. PMID: 38212419.
Chen X, Le J, Hu Y. Predicting Schistosomiasis Intensity in Africa: A Machine Learning Approach to Evaluate the Progress of WHO Roadmap 2030. Am J Trop Med Hyg. 2024 May 21:tpmd230751. doi: 10.4269/ajtmh.23-0751. Epub ahead of print. PMID: 38772355.
Lamberti O, Kayuni S, Kumwenda D, Ngwira B, Singh V, Moktali V, Dhanani N, Wessels E, Van Lieshout L, Fleming FM, Mzilahowa T, Bustinduy AL. Female genital schistosomiasis burden and risk factors in two endemic areas in Malawi nested in the Morbidity Operational Research for Bilharziasis Implementation Decisions (MORBID) cross-sectional study. PLoS Negl Trop Dis. 2024 May 8;18(5):e0012102. doi: 10.1371/journal.pntd.0012102. PMID: 38718065; PMCID: PMC11104661.
Anisuzzaman, Hossain MS, Hatta T, Labony SS, Kwofie KD, Kawada H, Tsuji N, Alim MA (2023) Food- and vector-borne parasitic zoonoses: global burden and impacts. Advances in Parasitology. 120:87-136.
Anisuzzaman and Tsuji N (2020) Schistosomiasis and hookworm infection in humans: Disease burden, pathobiology and anthelmintic vaccines. Parasitology International. 75:102051.
Labony SS, Hossain MS, Hatta T, Dey AR, Mohanta UK, Islam A, Shahiduzzaman M, Hasan MM, Alim MA, Naotoshi Tsuji N, Anisuzzaman (2022) Mammalian and Avian Larval Schistosomatids in Bangladesh: Molecular Characterization, Epidemiology, Molluscan Vectors, and Occurrence of Human Cercarial Dermatitis. Pathogens, 11:1213.
Anisuzzaman, Frahm S, Prodjinotho UF, Bhattacharjee S, Verschoor A, da Costa CP (2021) Host-specific serum factors control the development and survival of Schistosoma mansoni. Frontiers in Immunology, 12:635622.
Frahm S, Anisuzzaman A, Prodjinotho UF, Vejzagić N, Verschoor A, Prazeres da Costa C. 2019 A novel cell-free method to culture Schistosoma mansoni from cercariae to juvenile worm stages for in vitro drug testing. PLoS Neglected Tropical Diseases. 13(1):e0006590.
Minor
Make the scientific name italic
Do not start sentence with abbreviations
Too many paragraphs, make them meaningful by merging
Comments on the Quality of English Language
Minor
Author Response
In attachment please find our reply to reviewer 2.

Reviewer 3 Report
Comments and Suggestions for Authors
Dear Authors,
Please find an attached file.
Thank you.

Author Response
In attachment please find our response to Reviewer 3.

Round 2
Reviewer 1 Report
Comments and Suggestions for Authors
Dear authors,
Thank you for addressing the comments given point by point. I have reviewed the revised version of the manuscript. I have no further comments. I wish you all the best in your ongoing and future research work on schistosomiasis.
Author Response
Thank you for accepting our manuscript.
Reviewer 2 Report
Comments and Suggestions for Authors
The manuscript can be accepted for publication now.
Comments on the Quality of English LanguageMinor
Author Response
Thank you for accepting our manuscript.
Reviewer 3 Report
Comments and Suggestions for Authors
Dear Authors,
Please find an attached file.

Author Response
Please find the response to reviewer's comments in attachment.

Round 3
Reviewer 3 Report
Comments and Suggestions for Authors
Dear authors,
Please see an attached file.

Author Response
Dear Reviewer,
In attachment please find the last version of our manuscript after removing the part of S. Mansoni

Round 4
Reviewer 3 Report
Comments and Suggestions for Authors
Dear Authors,
Thank you very much for taking my suggestions in account.
I don't have any objections or concerns for the manuscript.
Thank you again for your great efforts.